# Improving Mechanical Properties of Tendon Allograft through Rehydration Strategies: An In Vitro Study

**DOI:** 10.3390/bioengineering10060641

**Published:** 2023-05-25

**Authors:** Chun Bi, Andrew R. Thoreson, Chunfeng Zhao

**Affiliations:** 1Orthopedic Biomechanics Research Laboratory, Department of Orthopedic Surgery, Mayo Clinic, 200 First Street SW, Rochester, MN 55905, USA; doctorbichun@126.com; 2Orthopaedic Traumatology, Trauma Center, Shanghai General Hospital, School of Medicine, Shanghai Jiao Tong University, 650 Xin Songjiang Road, Shanghai 201620, China; 3Materials and Structural Testing Core Laboratory, Mayo Clinic, 200 First ST SW, Rochester, MN 55905, USA

**Keywords:** tendon, allograft, biomechanical properties, lyophilization, rehydrated solutions

## Abstract

Allogenic tendons grafts sourced from intrasynovial tendons are often used for tendon reconstruction. Processing is achieved through repetitive freeze–thaw cycles followed by lyophilization. Soaking the lyophilized tendon in saline (0.9%) for 24 h is the standard practice for rehydration. However, data supporting saline rehydration over the use of other hydrating solutions are scant. The purpose of the current study was to compare the effects of different rehydration solutions on biomechanical properties of lyophilized tendon allograft. A total of 36 canine flexor digitorum profundus tendons were collected, five freeze–thaw cycles followed by lyophilization were performed for processing, and then divided into three groups rehydrated with either saline solution (0.9%), phosphate-buffered saline (PBS), or minimum essential medium (MEM). Flexural stiffness, tensile stiffness, and gliding friction were evaluated before and after allograft processing. The flexural moduli in both fibrous and fibrocartilaginous regions of the tendons were measured. After lyophilization and reconstitution, the flexural moduli of both the fibrocartilaginous and non-fibrocartilaginous regions of the tendons increase significantly in the saline and MEM groups (*p* < 0.05). Compared to the saline and MEM groups, the flexural moduli of the fibrocartilaginous and non-fibrocartilaginous regions of tendons rehydrated with PBS are significantly lower (*p* < 0.05). Tensile moduli of rehydrated tendons are significantly lower than those of fresh tendons for all groups (*p* < 0.05). The gliding friction of rehydrated tendons is significantly higher than that of fresh tendons in all groups (*p* < 0.05). There is no significant difference in either tensile moduli or gliding friction between tendons treated with different rehydration solutions. These results demonstrate that allograft reconstitution can be optimized through careful selection of hydrating solution and that PBS could be a better choice as the impact on flexural properties is lower.

## 1. Introduction

As one of the most common hand injuries, flexor tendon injuries remain a clinical challenge for functional reconstruction [1,2,3,4]. Intrasynovial tendon allografts are quite useful for tendon repair due to the smooth surface, low friction, and durability and has been widely used for effective reconstruction of injured tendons [5,6,7]. However, the fresh allograft may induce an immune reaction. Processing of allograft tissue eliminates its immunogenicity and preserves it to be ready for use as needed. A common process for allograft preservation is to apply repetitive freeze–thaw cycles followed by lyophilization [8,9,10,11,12,13]. Before transplantation, rehydration of the allograft is essential [8,9,10]. Rehydration is typically achieved by soaking the allograft in physiologic saline (0.9%, PH 5.5 (4.5–7.0)) for 24 h [14].

Tensile and friction properties of flexor tendons are the most widely investigated mechanical properties. However, in order to traverse through the pulley of a flexing digit, the tendon’s ability to bend, which is characterized with a flexural stiffness in a previous study, is also of great importance and impacts its gliding resistance [15]. There are two different regions in an intrasynovial tendon. On the basis of their different composition, one is named the fibrocartilaginous region and the other is named the non-fibrocartilaginous region. Previous studies reveal that these separate regions exhibit different mechanical properties [16]. A previous study also show that after lyophilization and rehydration with saline (0.9%), the flexural moduli of these two regions significantly increases, which may have negative effects on the clinical utility of a reconstituted tendon allograft in vivo [15]. Other studies showed that after rehydration with saline (0.9%), the gliding resistance of tendon was increased [14,17]. However, it must be noted that all of these studies reported using saline (0.9%) as the rehydration solution. In light of this, it cannot be ruled out that the selection of rehydration solution may impact the mechanical properties of a tendon allograft. Further engineering of the reconstitution process may result in mechanical properties that are more desirable for clinical flexor tendon repair.

Phosphate-buffered saline (PBS, PH 7.4), because it is isotonic and non-toxic to most cell types, is a solution extensively applied in biological research for substance dilution and cell container rinsing. The salts dissolved into PBS include NaCl, KCl, Na_2_HPO_4_, and KH_2_PO_4_. The osmolarity and ion concentrations of the PBS are very close to those of the human body, making it a potentially good candidate solution for tendon rehydration [18,19,20]. As a cell culture medium, minimum essential medium (MEM, PH 5.1–5.5 (10×)) is widely used to maintain cells in tissue culture. MEM includes a number of soluble components that include vitamins, glucose, phenol red, iron, amino acids, and salts. The salts dissolved into MEM include NaCl, MgSO_4_, CaCl_2_, and NaH_2_PO_4_. It is suitable to sustain molecular processes and growth for most types of cells and has been shown to have beneficial effects on tissue after transplantation in vivo [21,22,23]. In view of the isotonicity and osmolarity of PBS and the beneficial effects of MEM imparted in tendon transplantation applications, it is reasonable to propose that these or other rehydration solutions may be potential alternatives to saline that may result in a superior mechanical performance to saline, perhaps even improving tendon mechanical properties instead of degrading them as has been shown with saline reconstitution. This study aimed to compare the effects of different rehydration solutions on the mechanical properties of a lyophilized tendon allograft. We hypothesized that the mechanical properties of rehydrated tendon allograft will differ with the selection of different rehydration solutions.

## 2. Materials and Methods

***Specimens:*** Twelve hind paws were harvested from six adult mongrel dogs aged approximately 1 year. These dogs were sacrificed for other, unrelated Institutional Animal Care and Use Committee (IACUC)-approved studies; treatments applied in these studies were not anticipated to have had any structural or mechanical effect on hind limb musculoskeletal tissues. The hind paws were wrapped in saline-soaked gauze and stored in a -80 °C freezer immediately after sacrifice for future testing. Specimens were thawed at room temperature in a water bath, and the flexor digitorum profundus (FDP) tendons (intrasynovial) from the 2nd, 3rd, and 4th digits were harvested from each paw, accumulating a total of 36 tendons. Previous studies show there are no anatomical, histological, or biomechanical differences between tendons from these digits [24].

***Tendon preparation***: FDP tendons were cut into segments all of equal length that included the fibrocartilaginous (fc) and non-fibrocartilaginous (nfc) regions. The tendon dimeters along the major and minor axes of all 36 tendons were measured at the same location, repeating the measurements three times using a digital caliper (Digimatic Absolute, Mitutoyo, Aurora, IL); repeated measures of each diameter were then averaged. Before lyophilization, assessments of flexural, tensile, and friction properties were performed to characterize the baseline untreated mechanical properties for all 36 tendons. Mechanical testing applied was non-destructive to permit pair-wise evaluation of property changes between two different time points. Tendons were then processed into allograft tissue through techniques that are known to reduce immunogenicity. This process included 5 cycles of flash-freezing and thawing of the tissue. Each tendon was immersed in a liquid nitrogen bath for 1 min. After that, tendons were thawed in 37°C saline (0.9%) for 5 min. After 5 repeated freeze–thaw cycles, the lyophilization of all tendons were performed for 12 h using a laboratory freeze-drying system (Benchtop Manifold Freeze Dryer, Millrock Technology, Kingston, NY). Lyophilized tendons were then randomly divided equally into three groups, differentiated by the solution to be used for rehydration to be: group (A), saline (0.9%); group (B), phosphate-buffered saline (PBS); and group (C), minimum essential medium (MEM). Regardless of solution used for rehydration, tendons were rehydrated by soaking in solution at 4 °C for twenty-four hours. All mechanical property assessments, including flexural, tensile, and friction testing, were repeated immediately after rehydration.

***Flexural testing***: Flexural testing of both the fc and nfc regions was performed using a custom-fabricated apparatus designing to apply three-point bend loading using means described previously [15]. Briefly, the device consisted of a recessed well retaining a room-temperature saline bath, a simple support formed by two vertical steel pins, and an additional loading pin driven by a mechanical actuator and connected to a load cell (GSO-150, Transducer Techniques, Temecula, CA, USA), as well as linear position sensor (TR-50, Novotechnik, Southborough, MA, USA). The span between support pins was 12.5 mm, and three loading cycles were applied at a constant displacement rate of 0.05 mm/s to a maximum displacement of 2 mm, followed by a return to the starting point (Figure 1). The force and displacement data were recorded at a sample rate of 100 Hz. The data collected from the second cycle in the series were used to calculate flexural modulus. By placing a different region of the tendon over the support pins, the flexural properties of both the fc and nfc regions could be assessed with this method.

***Tensile testing***: The FDP tendon has a variable cross-section over its length. Therefore, to capture this variation in tensile calculations, each tendon was marked to create 8 equal sections over a 40 mm test length to create 5 mm segments. A digital caliper was used to measure the major and minor diameters at the boundaries of each segment. A previous study showed that there was not any structural damage, or any changes to mechanical properties, as long as tendons were not strained to more than 5% [25]. Therefore, by means of a servohydraulic test system (Figure 2) (858 Mini Bionix II, MTS Systems, Eden Prairie, MN, USA), each tendon was distracted to a maximum displacement of 2 mm at a constant rate of 20 mm/min. Force and displacement data were collected at a sample rate of 50 Hz. Due to limited specimen length and physical limitations in gripping tendon samples without damaging them, tensile property assessment could not be localized separately to the fc and nfc regions; the values obtained represent the average properties over the entire length.

***Friction testing***: Friction testing was conducted according to previously established methods [26]. Briefly, a four-mm-diameter glass rod was fixed to the drive shaft of a stepper motor, aligning their longitudinal axes. Each tendon was linked to a load cell with a string at one end, strung under the horizontally aligned rod, then attached to another string at the other end that was looped over a pulley and linked to a 200 g (1.96-N) weight. Between the horizontal plane and the string, an angle of 25° was set on both sides. The tendon and rod were immersed in a room-temperature saline bath. The rod was rotated clockwise with a surface velocity of 2 mm/s for 2 revolutions (equivalent to approximately 25 mm of linear excursion) and then similarly rotated counterclockwise for 2 revolutions, which was defined as a cycle (Figure 3A,B represent the over view and lateral view, respectively). Three cycles were applied. Load cell data were collected at a sample rate of 50 Hz. By readjusting the position of the tendon and bringing the rod in contact with either the fc or nfc region of the tenon, the properties of both regions were assessed.

***Data analysis***: Making the assumption that the cross-sections of the tendon were elliptical in shape, the tendon cross-sectional areas were calculated using the area equation for an ellipse. Flexural modulus (E) was calculated using relationships previously described based on classical equations for beam deflection [15]. Average tensile modulus, *E*, was calculated taking into consideration the varying cross-sections using Equation (1), where *k* is the stiffness obtained from the slope of the linear region of the force–displacement curve, *L* is the test length of the tendon, and *a_i_* and *b_i_* are the major and minor radius of each tendon section, respectively.
(1)$$E=kL\sum_{i=1}^{8}\frac{1}{\pi a_{i}b_{i}}$$

The average friction force was calculated by taking the average of the difference of the force acting in the direction of the excursion and the force in the opposing direction over the entire excursion cycle for the third cycle.

A sample size of 10 per group would allow detection of a difference, according to an a priori power analysis, between means of 1.3 standard deviations with 80% power and 95% confidence. Data were analyzed using SPSS 19.0, and expressed as the mean ± standard deviation. One-way ANOVA with post hoc contrasts by the Tukey–Kramer test was performed to compare the differences of flexural modulus, tensile modulus, and friction force among these three tendon groups in the nc and nfc regions. A *p*-value of 0.05 or less was considered statistically significant in all cases.

## 3. Results

The tendon cross-sectional area after lyophilization and rehydration with different solutions is significantly (*p* < 0.01) increased in both fc and nfc regions in all groups compared to the fresh (pre-lyophilization) tendons. However, there is no significant difference between different solution groups (Figure 4). Compared with pre-lyophilization (control) groups (increases 142.9% and 154.4% between pre- and post-saline rehydration in fc and nfc, respectively, and increases 139.9% and 130.9% between pre- and post-MEM rehydration in fc and nfc, respectively), the flexural moduli in both fc and nfc regions of FDP tendon in the saline and MEM rehydration tendon groups increase significantly (*p* < 0.05). However, there is no significant difference in flexural moduli before and after rehydration in the PBS group. Compared to the saline and MEM groups, the flexural moduli of the fc and nfc regions of tendons rehydrated with PBS are significantly lower (*p* < 0.05 for both) (Figure 5).

Compared with those of fresh tendon groups, tensile moduli in the all three groups are significantly lower (*p* < 0.05). The average tensile moduli of the saline, PBS, and MEM groups after rehydration are 71.6%, 69.1%, and 76.8% of those fresh tendon groups, respectively. However, no significant difference in tensile moduli is found between the three rehydration treatments (*p >* 0.05 for all) (Figure 6).

Compared with the fresh tendon groups, the friction force in all three groups is significantly higher (*p* < 0.05 for all) after lyophilization and rehydration. The average friction force of the saline, PBS, and MEM groups after rehydration is 161.4%, 141.2%, and 160.6% of those of the fresh tendon groups, respectively. No significant difference in frictional force is found among these three groups (*p >* 0.05 for all) (Figure 7).

## 4. Discussion

Accessibility to tendon allograft tissue provides an important option in the treatment of flexor tendon injury. The processing of allograft tendon tissue has been shown to negatively impact their mechanical properties in ways that could increase gliding resistance. However, there are a number of parameters in the production process that could be altered that may result in improved properties. The objective of this study was to explore the role one of these parameters, rehydrating solution, on resulting properties. Realizing this objective may not only introduce a processing option that may improve clinical outcomes, but it would also give cause to explore more options beyond those considered in this study to develop a superior allograft product.

In the current study, the flexural moduli of an intrasynovial tendon allograft in fc and nfc regions of its fresh and rehydrated, lyophilized state were investigated based on our previous testing method [15,17]. Flexural moduli can impact the ability of a tendon to wrap around the curvature of the pulley, which directly impacts gliding resistance. The flexural moduli of tendons rehydrated with saline and MEM are not fully restored to the normal tendon flexural modulus in either the fc region or the nfc region, leading to allograft tendons becoming stiffer than normal tendons. The changes in collagen cross-links resulting from the rehydration may relate to the increased flexural modulus observed. Saline or MEM may not able to break all of the collagen cross-linking that occurs as a result of lyophilization. However, the flexural modulus in PBS group is fully recovered to the pre-lyophilization status and is significantly lower compared to the saline and MEM groups. Previous studies identify that PBS causes tendon swelling, and induces microstructural changes to some extent. SEM images show that the collagen fibril diameter is increased to a certain degree while, at the same time, the fibril packing density decreases [27,28,29]. On the basis of these viewpoints, one possible explanation for the different performance of the PBS is that the isotonic properties and osmolarity, as well as some ingredients of PBS, cause more solution to penetrate into the lyophilized tendon or make some changes in tendon fiber structure that have not been characterized. Further histological analysis of rehydrated tendons needs to be performed to observe such effects, if present.

The tensile mechanical properties of rehydrated lyophilized tendons significantly decrease in all groups, regardless of rehydrating solution used. However, no significant differences are found in the tensile moduli among the three rehydration groups. According to previous studies, the osmolarity of PBS increases water content due to swelling, which decreases the tissue’s tensile modulus [30,31]. Based on these studies, we considered it is possible that with 24 h of rehydration, the number of fibrils per unit area decreases because fibril diameter and interfibrillar space increase. Such an effect would be in accordance with those of another study [32], in which it was shown the decrease in triple helix content and in collagen fibril density per mass unit could explain the decrease in ultimate tensile stress. Although the solution can penetrate the lyophilized tendon, the number of collagen fibrils that play the main role in the tensile testing decreased per mass unit, which cannot afford more strength during the tensile testing [28,33,34]. While tendon tensile properties may not directly impact tendon gliding, the sensitivity of this property to the selection of rehydration solution may reinforce observations in mechanical property and structural changes that have a more direct effect.

After rehydration, the friction force significantly increases in all three groups compared with the fresh tendon. The observation of the higher gliding resistance that results can be explained by the tendon surface morphology changes caused by freezing and lyophilization. The freezing and lyophilization may change the tendon surface morphology, resulting in the higher gliding resistance during the tendon motion [14,17]. However, we do not find any significant differences in the friction force among these three groups, although the mean friction force in the PBS group is observed to be lower compared to the saline and the MEM groups. Bloom et al. found that even with a physiological concentration, PBS decreases the tendon modulus and changes the microstructure of the tendon at some level [33]. Based on their studies, as well as the constitution and the PH of these solutions, we speculate that because of the isotonic properties and osmolarity of PBS, more PBS penetrates into the lyophilized tendon during the rehydration. As a result, the rehydrated tendon becomes softer. When the glass pin rolls on the tendon surface, the tendon surface more readily deforms, resulting in lower friction forces compared with the other two groups. The increased ease with which the tendon can deform when it curves may beneficially impact the gliding resistant between the rehydrated allograft tendon and a digit pulley, therefore, it can be said that the influence a rehydrating solution has on flexural stiffness and gliding resistance may be coupled.

The results presented from this study need to be interpreted in light of certain limitations. First, tendon size across different dogs was different, and we assumed that the cross-sectional of tendons were elliptical in shape at all intervals in our current study; in actuality, their shape may be more complex and varying along the tendon length. The accuracy of calculations of biomechanical properties may be affected by the simplification of the geometry. Second, the results were based on the biomechanical tests only. Further histological and cross-linking analyses of rehydrated, lyophilized tendon need to be performed in an attempt to establish a connection between changes in mechanical properties and alterations in tissue structure. Third, the results of the current study were based on a canine model in vitro and it would be affected by complicated conditions in vivo, and more relevant studies focused on human tissue should be taken for clinical application in the future. Additionally, while the tendons used on this study were known to have two distinct structural regions, it was not possible using the methods we employed to differentiate all mechanical properties between the fc and nfc regions. The fact that this study applies in vitro test methods to simulation mechanical performance of the tendon cannot be completed disregarded; in vivo mechanical performance may differ and it must also be noted that recellularization of the allograft may be impacted by the choice of rehydrating solution. Despite these potential limitations, these study results provide promising evidence that using PBS solution for rehydration of tendon allografts may provide better biomechanical behavior for lyophilized tendon allografts. We must also note that the options for rehydrating solution are not exhausted in this study and that there may be other solutions that could induce even less alteration to tendon mechanical properties.

## 5. Conclusions

For the fibrocartilaginous as well as the non-fibrocartilaginous regions of the tendons, the flexural moduli increase after lyophilization and reconstitution. Compared to the saline and MEM groups, the flexural moduli of the fibrocartilaginous and non-fibrocartilaginous regions of tendons rehydrated with PBS are significantly lower. Tensile moduli of rehydrated tendons are significantly lower and the gliding friction of rehydrated tendons is significantly higher for all tendons after lyophilization and rehydration, regardless of the rehydration solution used. These results demonstrate that allograft reconstitution can be optimized through careful selection of rehydrating solution and that PBS could be a better choice, as the impact on flexural properties is lower. 

## Figures and Tables

**Figure 1 bioengineering-10-00641-f001:**
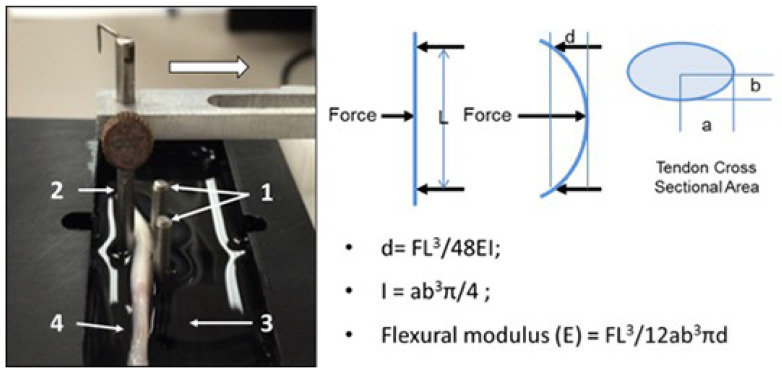
The left photo shows the flexural testing of both the fc and nfc tendon regions, which was performed using a custom-fabricated apparatus designed to apply three-point bending method. The sample tendon (4) was mounted in the device that consisted of a recessed well (3) that retained a room-temperature saline bath, two vertical pins that served as simple supports (1), an additional loading pin driven by a mechanical actuator and connected to a load cell (2), and linear position sensor. The span between support pins was 12.5 mm, and three loading cycles were applied at a constant rate of 0.05 mm/s to a displacement of 2 mm, followed by a return to the starting point. Force and displacement data were collected at a sample rate of 100 Hz. Data from the 2nd cycle were used for property calculations. The diagram shows the simplified diagram of the force loading the tendon and the cross-sectional area of tendon. L is the span between supports, d is the deflection of the tendon, a is the major axis diameter of the tendon, and b is the minor axis diameter of the tendon.

**Figure 2 bioengineering-10-00641-f002:**
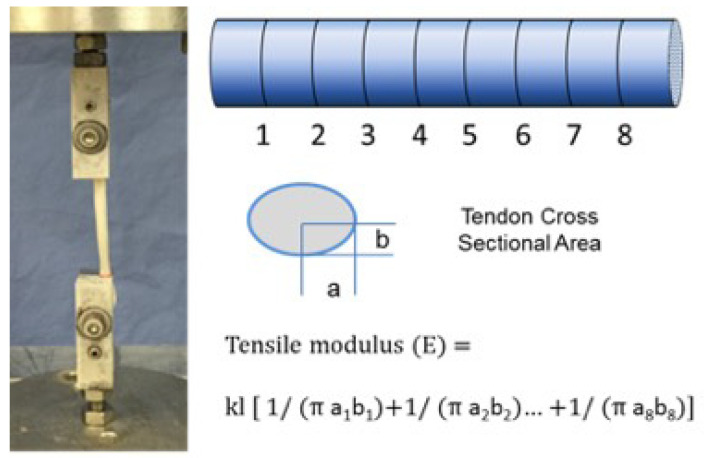
The diagram shows the tensile testing of tendon using a servohydraulic test system. Recording force and displacement at a sample rate of 50 Hz. As the FDP tendon has a variable cross-section over its length, the right photo shows that to capture this in tensile calculations, each tendon was marked to create 8 equal sections over a 40 mm test length to create 5 mm segments. The major (a1–a8) and minor diameters (b1–b8) at the boundaries of each segment were measured with a digital caliper.

**Figure 3 bioengineering-10-00641-f003:**
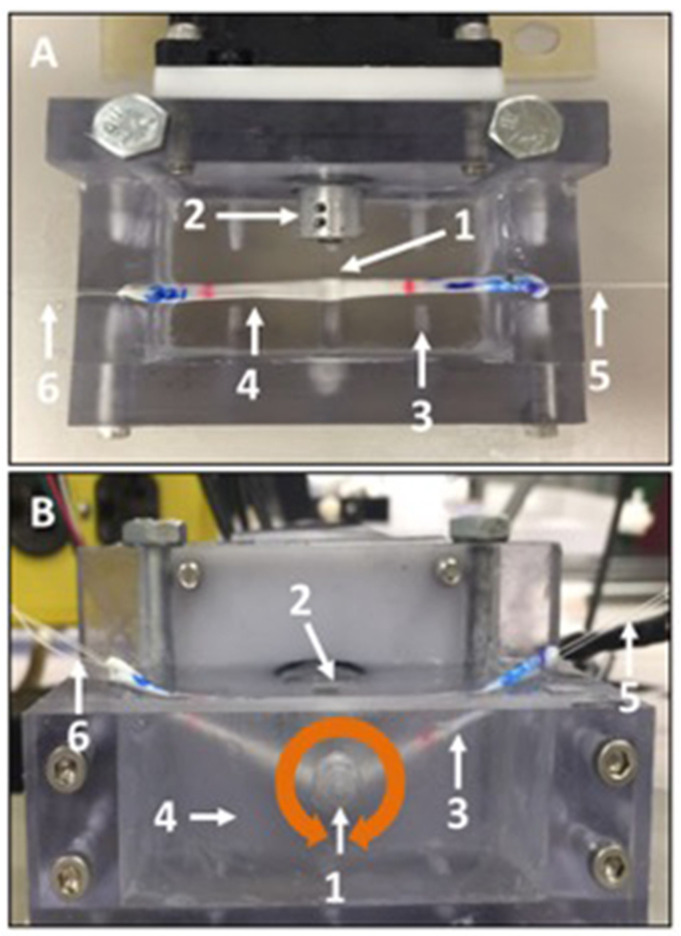
The photo shows the friction test. (**A**) shows top-down view of the device and (**B**) shows front-side view of the device. A four-mm-diameter glass rod (1) was connected to the drive shaft of a stepper motor (2), aligning their longitudinal axes. Each tendon was connected to a load cell (6) with a string at one end (4), strung under the rod with the horizon then attached to another string at the other end (3) that was strung over a pulley and connected to a 200 g (1.96-N) weight (5). An angle of 25° between the horizontal plane and the string was set on both sides. Tendons were immersed in saline. The rod was rotated clockwise with a surface velocity of 2 mm/s for 2 revolutions and then similarly rotated counterclockwise 2 revolutions, which was defined as a cycle. Three cycles were applied. Sampling force data at 50 Hz.

**Figure 4 bioengineering-10-00641-f004:**
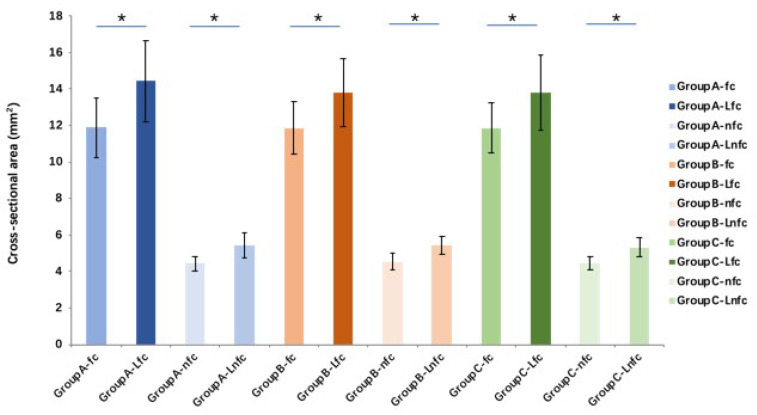
The tendon cross-sectional area after lyophilization and rehydration with different solutions in both fc and nfc regions in all groups (* indicates *p* < 0.01) (L means lyophilization and rehydration). Group A: saline (0.9%), group B: phosphate-buffered saline (PBS), or group C: minimum essential medium (MEM).

**Figure 5 bioengineering-10-00641-f005:**
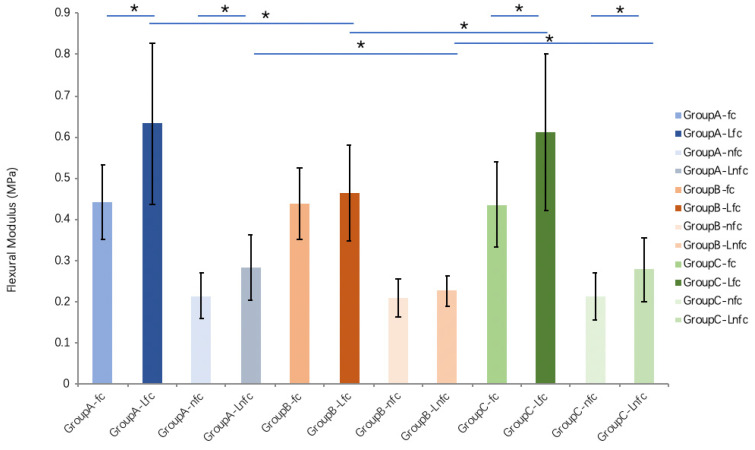
The flexural moduli in both fc and nfc regions of FDP tendon after lyophilization and rehydration with different solutions (* indicates *p* < 0.05) (L means lyophilization and rehydration). Group A: saline (0.9%), group B: phosphate-buffered saline (PBS), or group C: minimum essential medium (MEM).

**Figure 6 bioengineering-10-00641-f006:**
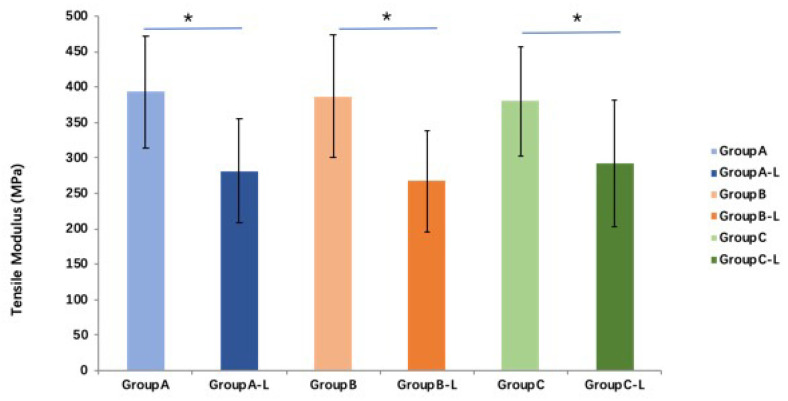
The tensile moduli of FDP tendon after lyophilization and rehydration with different solutions (* indicates *p <* 0.05) (L means lyophilization and rehydration). Group A: saline (0.9%), group B: phosphate-buffered saline (PBS), or group C: minimum essential medium (MEM).

**Figure 7 bioengineering-10-00641-f007:**
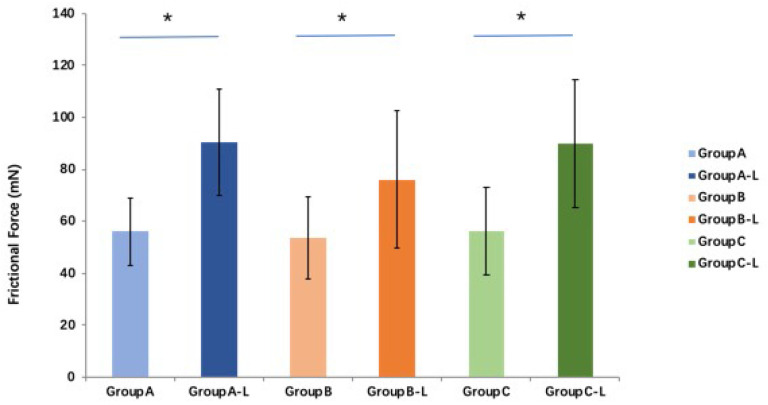
The friction force of FDP tendon after lyophilization and rehydration with different solutions (* indicates *p* < 0.05) (L means lyophilization and rehydration). Group A: saline (0.9%), group B: phosphate-buffered saline (PBS), or group C: minimum essential medium (MEM).

## Data Availability

Not applicable.

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
