# Peer review of "Improving Mechanical Properties of Tendon Allograft through Rehydration Strategies: An In Vitro Study"

_bioengineering, 2023, doi:10.3390/bioengineering10060641_

Round 1

Reviewer 1 Report

The Authors describe the effects of reydration methods on the properties of tendon allograft. The paper scientifically sounds and the results are convincing. However, it should be emphasized that these are dog tendons and therefore energy should be spent discussing the degree of similarity to human tendons. Indeed, histological differences can be often noted in the tissue organization among different species in mammals. Moreover, more details of samples could be added, such as the age of dogs and from which limbs the tendons have been taken (fore- hind-). One of the major limit in this work is the fact that 6 mongrel dogs had surely different body mass and their tendons had different in lenght. The figures are useful. The bibliography could be increased. 

Author Response

Reviewer 1

The Authors describe the effects of reydration methods on the properties of tendon allograft. The paper scientifically sounds and the results are convincing. However, it should be emphasized that these are dog tendons and therefore energy should be spent discussing the degree of similarity to human tendons. Indeed, histological differences can be often noted in the tissue organization among different species in mammals. Moreover, more details of samples could be added, such as the age of dogs and from which limbs the tendons have been taken (fore- hind-). One of the major limit in this work is the fact that 6 mongrel dogs had surely different body mass and their tendons had different in length. The figures are useful. The bibliography could be increased. 

Response: We thank the reviewer for the careful review and constructive comments. We have discussed the limitation of using canine tendon as a model as the following “Third, the current results were based on a canine model in vitro and it would be affected by complicated conditions in vivo; and it would be more meaningful to perform a study on human tissue in the future for clinical application.” (line 265). We have added dog age as suggested (Line 81). We used hind limb from each dog which has been clarified in the text (line 81). We agree with the reviewer that the tendon size across dogs were different in length and cross sectional area. However, we cut the tendon in a equal length (line 90) before testing, and the cross sectional area was normalize by calculating the flexural modulus (line 114 with an equation). Some relevant references have been added.

Reviewer 2 Report

In this manuscript, the authors compared the mechanical properties of tendon allografts after treatment with different rehydration solutions. In general, only some routine work got involved in this manuscript, and the explanations in the discussion part were not supported by the literature or the presented data. However, there are some specific comments for this manuscript:

1. For the discussion part, the authors should cite proper references to support their explanation of their data.

2. The authors did not define “Lnfc” which is shown in the figures.

3. The authors are suggested to indicate Group A, Group B, and Group C in the figure legends.

4. There are typos, such as two spaces on Page 3, Line 91.

Author Response

Reviewer 2

In this manuscript, the authors compared the mechanical properties of tendon allografts after treatment with different rehydration solutions. In general, only some routine work got involved in this manuscript, and the explanations in the discussion part were not supported by the literature or the presented data. However, there are some specific comments for this manuscript:

  1. For the discussion part, the authors should cite proper references to support their explanation of their data.

Response: Thanks for your comment. We have checked references that are properly cited in the manuscript.

  1. The authors did not define “Lnfc” which is shown in the figures.

Response: Thanks for your kindly comment, the explanations have been added in the figures.

  1. The authors are suggested to indicate Group A, Group B, and Group C in the figure legends.

Response: Thanks for your kindly suggestion. The Group A, B and C were indicated in figure legends.

  1. There are typos, such as two spaces on Page 3, Line 91.

Response: Thanks for your kindly comment. The typos have been deleted.

Reviewer 3 Report

A question relates to expand the exact differences between 0.9% saline and PBS solution.

Line 82-82: A question relates to the processing of the tissue following harvest, the authors state that paws were “stored in a -80°C freezer. Following thawing at room temperature in a water bath”, could they comment on any potential damage this may be doing to the tissue and could this be impacting the results observed?

Line 89: “All FDP tendons were cut into equal segments” are these equal lengths? To clarify

Line 91: “measured at the same location” which location? To clarify

Line 92: “Before lyophilization, assessment of flexural, tensile and friction properties was performed to characterize untreated mechanical properties for all 36 tendons.” Does this damage the tissue or change the mechanical properties prior to lyophilization?

Line 99: Could the authors comment on Group B: Phosphate- buffered saline (PBS), and the the exact differences between 0.9% saline and PBS solution.

Could all pH’s be states for each Group? (A-C)

Repetition in text for flexural loading and in the figure legend for figure 1, is not necessary

The device consisted of a recessed well that retained a room temperature saline bath, two vertical pins that served as simple supports and an additional loading pin driven by a mechanical actuator and 118 connected to a load cell and linear position sensor. The span between support pins was 12.5 mm and three loading cycles were applied at a constant rate of 0.05 mm/s to a displacement of 2 mm, followed by a return to the starting point. Force and displacement data were collected at a sample rate of 100 Hz. Data from the 2nd cycle was used for property calculations”

Line 122: “The right photo shows” there is no right photo, it is a diagram, To edit and clarify

Line 124: a Question relates to the “a is the major axis diameter of the tendon, and b is the minor axis diameter of the tendon.” As shown in figure one, could the authors comment on the oval shape of the tendon as opposed to a circular cylindrical structure. Is there evidence for this? If so could the authors support? Line 162: Why are the tendons assumed to be elliptical in shape? “Tendon cross-sectional areas were calculated assuming the sections were elliptical in shape.”

Line 139: “right photo” again not a photo, a diagram

Line 157: error in reference to figure “another string at the other end (3)” should it be (5)?

In Figure 3 the A and B are not used in the text or described in the figure legend. (Overview and lateral view) to amend.

Figure 4: Instead of Group A, B and C listed could the authors change labelling to either Saline, PBS or MEM. Fc and nfc need to be expanded in figure legend.

Have the authors compared the cross sectional area data between groups (A,B and C) for fc, nfc, Lfc and Lnfc? Data in Figure 4 suggests it is just a within group comparison, or perhaps there is no difference across groups? The data for each group also looks remarkably similar – and the variance the same, could the authors either present the raw data in supplementary and state what is being shown in the variance SD or SEM?

Could the authors comment on why Lyophilization is increasing CSA in all cases? Could it be to do with the method?

A suggestion to combine figure 4 and 5 together with parts A and B.

Blue lines showing significance in Figure 5 are hard to see and very close together, suggested to amend with lines with bracket ends.

Again in figure 5 what variance is being shown and how many replicates have been measured. The data for group A-fc, group B-fc and group C-fc look remarkable similar, could the author comment?

Line 202: suggested change of term from among to between.

Assumptions from the data presented in Figure 6 is that the fc and nfc could not both be tested for tensile properties, is this correct or is the data merged?

Could figures 6 and 7 be combined to one figure labelled A and B.

Line 225: spelling of collagen

Line 247- 249: Authors state “According to the constitution and the PH of these solutions, we speculated that because of the isotonic properties and osmolarity of PBS, more PBS penetrates into the lyophilized tendon during the rehydration. As a result, the rehydrated tendon became softer.” The pH of all solutions need to be stated in the methods section and more information about the contents of PBS should be mentioned, specific salts etc.

Author Response

Reviewer 3

A question relates to expand the exact differences between 0.9% saline and PBS solution.

Response: Thanks for your kindly comment. The differences were as follows:

Salts: PBS (NaCl, KCl, Na2HPO4, KH2PO4) VS  0.9% saline (NaCl)

pH: PBS (7.4) VS  0.9% saline (5.5 (4.5-7.0))

Others: PBS (Isotonic, Osmolarity) VS  0.9% saline (Isotonic)

Line 82-82: A question relates to the processing of the tissue following harvest, the authors state that paws were “stored in a -80°C freezer. Following thawing at room temperature in a water bath”, could they comment on any potential damage this may be doing to the tissue and could this be impacting the results observed?

Response: Thanks for your kindly comment.Actually, for tendon experiment, the routine operating process of paws was first stored in a -80°C freezer, then thawing at room temperature in a water bath before experiment.

Line 89: “All FDP tendons were cut into equal segments” are these equal lengths? To clarify

Response: Thanks for your kindly comment.“All FDP tendons were cut into equal segments” means after processing, these segments with equal length that included fibrocartilaginous (fc) and non-fibrocartilaginous (nfc) parts were performed to test in order to ensure consistency.

Line 91: “measured at the same location” which location? To clarify

Response: Thanks for your kindly comment.The same location means the measurements were taken at the major and minor diameters sites for each tendon in order to avoid the error as possible.

Line 92: “Before lyophilization, assessment of flexural, tensile and friction properties was performed to characterize untreated mechanical properties for all 36 tendons.” Does this damage the tissue or change the mechanical properties prior to lyophilization?  

Response: Thanks for your kindly comment. We don’t know if the lyophilization procedure would damage the tendons. However, we do believe that our testing procedures including friction test, flexural test, and tensile test within 5% did not damage the tendon structure. Thank you for your comment, and we have clarified this important point in the text (line 131). 

Line 99: Could the authors comment on Group B: Phosphate- buffered saline (PBS), and the the exact differences between 0.9% saline and PBS solution.

Response: Thanks for your kindly comment. The differences between 0.9% saline and PBS solution were as follows:

Salts: PBS (NaCl, KCl, Na2HPO4, KH2PO4) VS  0.9% saline (NaCl)

pH: PBS (7.4) VS  0.9% saline (5.5 (4.5-7.0))

Others: PBS (Isotonic, Osmolarity) VS  0.9% saline (Isotonic)

Could all pH’s be states for each Group? (A-C)

Response: Thanks for your kindly comment. Group A: saline (0.9%) pH 7.4, Group B: Phosphate-buffered saline (PBS) pH 5.1-5.5 (10x) and Group C: Minimum essential medium (MEM) pH 5.5 (4.5-7.0).

Repetition in text for flexural loading and in the figure legend for figure 1, is not necessary

“The device consisted of a recessed well that retained a room temperature saline bath, two vertical pins that served as simple supports and an additional loading pin driven by a mechanical actuator and 118 connected to a load cell and linear position sensor. The span between support pins was 12.5 mm and three loading cycles were applied at a constant rate of 0.05 mm/s to a displacement of 2 mm, followed by a return to the starting point. Force and displacement data were collected at a sample rate of 100 Hz. Data from the 2nd cycle was used for property calculations”

Response: Thanks for your kindly suggestion, the figure 1 legend has been simplifed.

Line 122: “The right photo shows” there is no right photo, it is a diagram, To edit and clarify

Response: Thanks for your kindly comment. It should be diagram, not photo, which has been edited in the manuscript.

Line 124: a Question relates to the “a is the major axis diameter of the tendon, and b is the minor axis diameter of the tendon.” As shown in figure one, could the authors comment on the oval shape of the tendon as opposed to a circular cylindrical structure. Is there evidence for this? If so could the authors support? Line 162: Why are the tendons assumed to be elliptical in shape? “Tendon cross-sectional areas were calculated assuming the sections were elliptical in shape.”

Response: Thank you for your thorough review. As Figure 1 indicates, "diameter" should be replaced with "radius". We have corrected the mistake: "a" should represent the major axis radius, while "b" should represent the minor axis radius of the tendon. Elliptical shape is the most commonly used method for calculating flexor tendon, rather than the circular cylindrical shape, as its anatomic features present a long axis and a short axis in its cross-sectional area.

Line 139: “right photo” again not a photo, a diagram

Response: Thanks for your kindly comment. It should be diagram, not photo, which has been edited in the manuscript.

Line 157: error in reference to figure “another string at the other end (3)” should it be (5)?

Response:  Thanks for your kindly comment. Actually, (3) and (4) represented the two ends of the tendon. The (6) represented a load cell and the (5) represented a 200-g (1.96-N) weight.

In Figure 3 the A and B are not used in the text or described in the figure legend. (Overview and lateral view) to amend.

Response:  Thanks for your kindly comment. The description of A and B of Figure 3 was added in the text.

Figure 4: Instead of Group A, B and C listed could the authors change labelling to either Saline, PBS or MEM. Fc and nfc need to be expanded in figure legend.

Response: Thanks for your kindly comment. The labelling of Group A, B and C was just for simplification as there were so many groups including fc and nfc regions in each tendon, and more groups were presented for comparison. Thus, in order to distinguish each group better, A, B and C were listed.

Have the authors compared the cross sectional area data between groups (A,B and C) for fc, nfc, Lfc and Lnfc? Data in Figure 4 suggests it is just a within group comparison, or perhaps there is no difference across groups? The data for each group also looks remarkably similar – and the variance the same, could the authors either present the raw data in supplementary and state what is being shown in the variance SD or SEM?

Response: Thanks for your kindly comment. Actually, only the within group comparison was taken to compare the change of the cross sectional area. The data for each group also looks remarkably similar. As these tendons were collected from 6 dogs. In order to avoid the deviation, totally 36 tendons were divided into 3 groups. By means of samples collection, the tendons of each group were collected from different dogs, and the numbers of tendon from each dog were the same in each group. Thus, mechanical properties of the fc and nfc part in each group before lyophilization and rehydration were similar. The raw data of the cross sectional area has been presented in supplementary.

Could the authors comment on why Lyophilization is increasing CSA in all cases? Could it be to do with the method?

Response: Thanks for your kindly comment. In our previous experiments, after lyophilization and rehydration with saline, we usually found the tendons changed, became stiffer and thicker, compared with their fresh status. Thicker could be deemed to cross sectional area (CSA) increased. Our previous published paper described this phenomenon. Thus, we try to find some methods to reduce the CSA as possible. And the different solution for rehydration were tested in this manuscript.

A suggestion to combine figure 4 and 5 together with parts A and B.

Response: Thanks for your kindly suggestion. Figure 4 and 5 represent the different data, cross sectional area and flexural moduli, respectively. Besides, the flexural moduli represented the mechanical property. The figure 4 just represented the area. It would not be suitable for combination of these two figures.

Blue lines showing significance in Figure 5 are hard to see and very close together, suggested to amend with lines with bracket ends.

Response: Thanks for your kindly suggestion. According to your suggestion, the positions of blue lines showing significance have been adjusted, in order to better clarification.

Again in figure 5 what variance is being shown and how many replicates have been measured. The data for group A-fc, group B-fc and group C-fc look remarkable similar, could the author comment?

Response: The flexural modulus of group A-fc, group B-fc and group C-fc, actually, were similar. As these tendons were collected from 6 dogs. In order to avoid the deviation, totally 36 tendons were divided into 3 groups. By means of samples collection, the tendons of each group were collected from different dogs, and the numbers of tendon from each dog were the same in each group. Thus, mechanical properties of the fc and nfc part in each group before lyophilization and rehydration were similar.

Line 202: suggested change of term from among to between.

Response: Thanks for your comment, it has been changed in the manuscript.

Assumptions from the data presented in Figure 6 is that the fc and nfc could not both be tested for tensile properties, is this correct or is the data merged?

Response: Thanks for your kindly comment. In the tensile testing, the PDP tendon including fc and nfc parts was regarded as a whole for test. Then was created into 8 equal sections for data processing. And the final data was the merged data.

Could figures 6 and 7 be combined to one figure labelled A and B.

Response: Thanks for your suggestion. The tensile moduli and the friction force, as two different mechanical properties, these two tests were taken with different test. So, it would be better displayed with two figures to show the differences of these two different properties.

Line 225: spelling of collagen

Response: Thanks for your comment, the word has been revised.

Line 247- 249: Authors state “According to the constitution and the PH of these solutions, we speculated that because of the isotonic properties and osmolarity of PBS, more PBS penetrates into the lyophilized tendon during the rehydration. As a result, the rehydrated tendon became softer.” The pH of all solutions need to be stated in the methods section and more information about the contents of PBS should be mentioned, specific salts etc.

Response:  Thanks for your suggestion, the pH of all solutions have been stated in the introduction. And the contents of PBS including the specific salts have been added in the manuscript.

Round 2

Reviewer 2 Report

The authors have addressed my concerns.

Author Response

We have provided point-by-point to the reviewer's comments below:

Reviewer 2

In this manuscript, the authors compared the mechanical properties of tendon allografts after treatment with different rehydration solutions. In general, only some routine work got involved in this manuscript, and the explanations in the discussion part were not supported by the literature or the presented data. However, there are some specific comments for this manuscript:

  1. For the discussion part, the authors should cite proper references to support their explanation of their data.

Response: Thanks for your comment. We have checked references that are properly cited in the manuscript.

  1. The authors did not define “Lnfc” which is shown in the figures.

Response: Thanks for your kindly comment, the explanations have been added in the figures.

  1. The authors are suggested to indicate Group A, Group B, and Group C in the figure legends.

Response: Thanks for your kindly suggestion. The Group A, B and C were indicated in figure legends.

  1. There are typos, such as two spaces on Page 3, Line 91.

Response: Thanks for your kindly comment. The typos have been deleted.

Reviewer 3 Report

Line 82-82: A question relates to the processing of the tissue following harvest, the authors state that paws were “stored in a -80°C freezer. Following thawing at room temperature in a water bath”, could they comment on any potential damage this may be doing to the tissue and could this be impacting the results observed?

Response: Thanks for your kindly comment. Actually, for tendon experiment, the routine operating process of paws was first stored in a -80°C freezer, then thawing at room temperature in a water bath before experiment.

Reviewer response: While I appreciate the routine opterating process, the authors have not address the comment sufficiently. I kindly repeat the question “Following thawing at room temperature in a water bath”, could they comment on any potential damage this may be doing to the tissue and could this be impacting the results observed?”

Line 92: “Before lyophilization, assessment of flexural, tensile and friction properties was performed to characterize untreated mechanical properties for all 36 tendons.” Does this damage the tissue or change the mechanical properties prior to lyophilization?  

Response: Thanks for your kindly comment. We don’t know if the lyophilization procedure would damage the tendons. However, we do believe that our testing procedures including friction test, flexural test, and tensile test within 5% did not damage the tendon structure. Thank you for your comment, and we have clarified this important point in the text (line 131). 

Reviewer response: this is not clear on line 131 of the current edited manuscript pdf

Repetition in text for flexural loading and in the figure legend for figure 1, is not necessary

“The device consisted of a recessed well that retained a room temperature saline bath, two vertical pins that served as simple supports and an additional loading pin driven by a mechanical actuator and 118 connected to a load cell and linear position sensor. The span between support pins was 12.5 mm and three loading cycles were applied at a constant rate of 0.05 mm/s to a displacement of 2 mm, followed by a return to the starting point. Force and displacement data were collected at a sample rate of 100 Hz. Data from the 2nd cycle was used for property calculations”

Response: Thanks for your kindly suggestion, the figure 1 legend has been simplifed.

Reviewer response: Figure 1 legend has not been simplified

Author Response

we have responded all reviewers' comments (see attached)
